# Optimization and Potentials of Kraft Lignin Hydrolysates Obtained by Subcritical Water at Moderate Temperatures

**Jaroslava Švarc-Gajić [1],\***, **Tanja Brezo-Borjan [1]**, **Richard J. A. Gosselink [2]**, **Ted M. Slaghek [2]**,
**Daniela Šojić-Merkulov [3]**, **Tamara Ivetić [3]**, **Szabolcs Bognár [3]** and **Zorica Stojanović [1]**

1 Faculty of Technology, University of Novi Sad, Bulevar Cara Lazara 1, 21000 Novi Sad, Serbia
2 Wageningen Food & Biobased Research, Bornse Weilanden 9, 6708 WG Wageningen, The Netherlands
3 Faculty of Sciences, University of Novi Sad, Trg Dositeja Obradovića 3, 21000 Novi Sad, Serbia
\* Correspondence: jsgajic@gmail.com; Tel.: +381-214853661

**Abstract:** Kraft lignin was treated with subcritical water at moderate temperatures (120–220 °C) in different gas atmospheres, with the goal of optimizing its depolymerization under mild conditions. Lignin depolymerization was observed and compared using different homogeneous and heterogeneous catalysts in both nitrogen and carbon dioxide atmospheres. The most important treatment parameters for maximum lignin depolymerization and the highest yields of phenolic and other aromatic monomers were optimized. The influence of the process temperature, pressure, and time in both gas atmospheres was defined and optimized for maximum liberation of monomers into the aqueous phase. The yields of total phenols and other aromatics in the nitrogen atmosphere were the highest at 150 °C, whereas treatment in the carbon dioxide atmosphere required higher temperatures (200 °C) for a comparable efficiency. The effects of phenol addition as a capping agent in lignin depolymerization were observed and defined for both gas atmospheres. Phenol addition caused a remarkable increase in the total phenols content in the aqueous phase; however, it did not significantly affect the contents of other aromatics. The antioxidant properties of lignin hydrolysates obtained at different temperatures in different gas atmospheres were compared, correlated with the total phenols contents, and discussed, showing the promising potential of lignin hydrolysates obtained under mild subcritical water conditions.

**Keywords:** kraft lignin; subcritical water; phenolic monomers; antioxidant properties





## 1. Introduction

Lignocellulose biomass is generated abundantly worldwide by the forestry and agri-food industry and traditionally, this waste is used for energy production and as a component for composite materials for animal bedding and soil amendments. Lignin, cellulose, and hemicellulose are the three main fractions in lignocellulose biomass and can be converted to valuable chemicals or fuel by different chemical, biochemical, thermochemical, and other technologies.

Lately, research has been focused on finding new pathways for lignin valorization. Instead of its use as a relatively inefficient burning fuel, more economical ways implying its conversion to biomaterials, biochemicals, biofuels, biopolymers, and other added-value commodities, such as vanillin, emulsifiers, phenolic resins, and a precursor for carbon fibers, are being developed.

Industrial applications of lignin imply its use as a binder in laminated and composite wood products, as asphalt binder, for interior panels production, as an adhesive for linoleum, or, more recently, for lignin oils production, which can further be processed to different biofuels [1,2]. Different lignin-based polymers and thermoplastics are being developed to replace fossil-based products and improve their biodegradability. Recent interest in lignin valorization has been directed towards the production of added-value

chemicals and platform molecules by different technologies, such as enzymatic conversion, hydrogenolysis, pyrolysis, solvolysis, or catalytic oxidative depolymerization.

Hydrothermal liquefaction of lignin implies its depolymerization in sub- or supercritical water. Supercritical water is water heated to temperatures above its critical 374 °C and maintained in a liquid state by the application of pressures. In processes with sub- and supercritical water, the temperature is the principle operational parameter. In hydrothermal decomposition, the reactivity of water increases dramatically, potentiating numerous decomposition reactions such as hydrolysis, dealkylation, decarboxylation, elimination hydrogenation/dehydrogenation, oxidation, etc. [3,4]. Potentiated reactivity is due to endothermic dissociation of water and an increase in water ion products. The ion product of water reaches its maximum at 200 °C [5], favoring various acid- or base-catalyzed reactions. The density and viscosity of water also decrease with heating, contributing to better contact of the reactive medium with the solid matrix. Taking into consideration the drop in water polarity with heating, its increased reactivity, and change in the other physico-chemical properties of water, by adequately choosing the operating temperature, the reaction conditions can be finely tuned for a specific purpose. The green character of sub-/supercritical water, low price, and high tunable reactivity make this technology very attractive in valorization of biowaste of both plant and animal origins, producing added-value compounds and platform molecules from biomass. Regarding lignin, most of the research has been directed towards the production of bio-oil. Belkheiri et al. [6] investigated the influence of different catalysts ($ZrO_2$, $K_2CO_3$/KOH, $Na_2CO_3$/NaOH) for lignin degradation in subcritical water (350 °C, 25 MPa), adding phenol as a capping agent. The authors analyzed the obtained bio-oil and aqueous fractions, separating them by centrifugation. The most dominant degradation phenolic compounds anisoles (0.6%), guaiacols (3%), and alkyl phenols (3.5%) were mostly represented in the bio-oil fraction. Their corresponding yields in the aqueous phase were 0.6% for anisoles, 1.6% for guaiacols, and alkyl 2.3% for phenols, with the exception of catechols, which showed a higher yield (6.1%) in the aqueous phase.

Islam et al. [7] studied the yield of different phenolic monomers in bio-crude obtained by subcritical water. The authors separated crude-oil and light-oil fractions from the aqueous phase by L/L extraction with acetone and ethyl acetate, respectively. The highest yield of bio-oil (44.5%) was obtained at 350 °C. In light-oil, many phenolic monomers, with a total yield of 1.2–6.3%, were detected in contrast to crude-oil, in which their yield was negligible. Belkheiri et al. [6] used different catalysts ($ZrO_2$, $K_2CO_3$, KOH) in the depolymerization of Kraft lignin, adding 2–10% of phenol as a capping agent. Phenol was shown to be effective even at low concentrations and did not the yield of bio-oil. Alkylated phenolic monomers increased with phenol addition, whereas aromatic compounds showed decreasing trends. Balawanthrao [8] studied the effects of different catalysts for alkali lignin depolymerization in subcritical water and concluded that the highest yield of phenolic monomers (40.85%) was obtained with Ni-Graphene catalyst at 240 °C and a 10 min reaction time.

Dell'Orco et al. [9] observed the effects of lignin depolymerization in $CO_2$ atmosphere in addition to investigating the influence of homogeneous catalysis (KOH). The addition of KOH intensified lignin depolymerization, but did not affect the yield of phenolic monomers.

The potential of other solvents for lignin depolymerization at high temperatures was tested. Ivakhnov et al. [10] compared the production of gaseous and liquid products of lignin decomposition at 450 °C (4 h) in supercritical methanol, ethanol, isopropanol, acetic acid, and their 50% solution. Isopropanol was shown to be the most stable at the tested temperatures, whereas acetic acid was completely degraded. The addition of water increased the stability of all tested solvents.

Most of the published papers dealing with hydrothermal liquefaction of lignin focused on the production and characterization of bio-oil fraction, and optimization of the operational parameters for maximizing the bio-oil yield. Only a few publications have dealt with the analysis of the aqueous phase remaining after lignin hydrothermal treatment and the

content of water-soluble phenolic monomers. The purpose of this work was to study lignin decomposition under mild reaction conditions using subcritical water and to compare the efficiency of hydrothermal lignin degradation under different conditions. Thus, Kraft lignin was treated with subcritical water in the temperature range 120–220 °C under nitrogen and carbon dioxide atmospheres. These conditions are much milder compared to the supercritical process of lignin treatment, which is typically conducted at temperatures above 374 °C. The effects of different catalysts, both homogeneous and heterogeneous, on the content of phenolic and aromatic monomers in the aqueous phase and the influence of phenol addition, as a capping agent, were studied. In the aqueous phase obtained by lignin treatment with subcritical water, the content of total phenols and other aromatics was calculated. In the thus obtained extracts, the total antioxidant activity was measured in order to evaluate the potential of this biowaste-derived product for further applications such as coatings or packaging materials.

## 2. Materials and Methods

### 2.1. Samples

Kraft lignin sample (Indulin AT), produced by Ingevity US, was obtained from the research institute Wageningen Food & Biobased Research, Wageningen, the Netherlands.

### 2.2. Chemicals and Reagents

Gallic acid and rutin trihydrate were purchased from Dr Ehrenstorfer GmbH (Augsburg, Germany). Aluminum chloride hexahydrate, 2,2′-azino-bis(3-ethylbenzothiazoline-6-sulfonic acid) and sodium carbonate were obtained from Alfa Aesar GmbH & Co KG (Karlsruhe, Germany). Folin–Ciocalteu reagent and potassium peroxodisulfate were purchased from Lachner (Neratovice, Chech Republic). Sodium hydroxide was acquired from Centrohem (Stara Pazova, Serbia), whereas hydrochloric acid and phenol were obtained from Zorka (Šabac, Serbia). The used zeolite (60 μm) was provided by Zeoworld doo, Serbia.

Titanium dioxide was provided by Comcen doo (Serbia), whereas the other tested commercial heterogenous catalysts ($ZrO_2$ and $CeO_2$) were provided by Alfa Aesar (Kandel, Germany) and Sigma-Aldrich (St. Louis, MO, USA), respectively. On the other hand, the newly synthesized cerium-zirconium oxide nanoparticles with a molar ratio of $CeO_2$:$ZrO_2$ = 80:20 were prepared by a two-step mechanochemical procedure. First, the commercial powders of $CeO_2$ and $ZrO_2$ were wet-milled using ethanol in a ball mill (Retsch PM100) with a zirconia vial (50 mL) and zirconia balls with a 3 mm diameter and a standard ball-to-powder weight ratio of 10:1 for 240 min. After the successful milling process, the obtained powder mixture was annealed at 700 °C for 120 min. All other chemicals were of analytical reagent grade.

### 2.3. Hydrothermal Treatment of Lignin

Lignin was treated with subcritical water in a home-made subcritical water reactor (Figure 1). Pressurized gases, nitrogen (99.999%) and carbon-dioxide (99.995%), were provided by Messer, Germany. The reaction vessel, with a total capacity of 1.7 L, was heated with a heating rate of approximately 10 °C/min. Agitation during the process was achieved by the movements of a vibrational platform with variable vibrating frequencies between 1 and 5 Hz.

Lignin samples and distilled water in a ratio of 1:30 were placed in the reaction vessel. In experiments with the capping agent and catalysts, 0.5% of phenol, 1% of homogeneous catalyst (NaOH or HCl), or 0.2% of heterogeneous catalyst (Zeolite, $TiO_2$, $ZrO_2$, $CeO_2$, $CeO_2$-$ZrO_2$ 80:20) were added to the reaction mixture. Homogeneous catalysts were dissolved previously in water. In the case of heterogeneous catalysts, the mixture was mixed and homogenized manually before the start of the experiment. After that, pressurization to different pressures of up to 40 bars with either nitrogen or carbon dioxide was carried out though the gas inlet built in the reaction vessel lid. The pressure was controlled by a

built-in manometer. The reaction vessel was then placed on a heating/vibrating platform and after the operational temperature (120–220 °C) was reached, the time (30–120 min) was measured. The operating temperature was maintained by a digital temperature controller. After the treatment, the process vessel was immediately cooled in a flow-through water bath at 20 ± 2 °C. Depressurization was carried out by valve opening and the purging of gases through a valve. The obtained aqueous fractions were separated by filtration through a Whatman filter paper, grade 1, and stored in a refrigerator at 4 °C for further analysis.

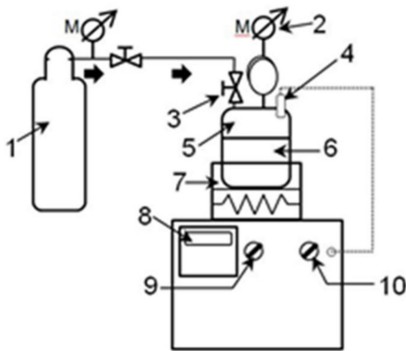

**Figure 1.** Schematic diagram of the subcritical water reactor: (1) gas cylinder; (2) manometer; (3) input gas valve; (4) thermocouple for temperature measurement; (5) coverlid of the reactor; (6) reaction vessel; (7) vibrating platform; (8) digital temperature controller; (9) main switch; (10) switch for the vibrating platform.

*2.4. Determination of Total Phenols Content*

The Folin–Ciocalteu method [11] was used to determine the total phenols content in the subcritical water fractions of lignin. The reaction mixture was prepared by mixing 0.1 mL of the extract, 7.9 mL of distilled water, 0.5 mL of Folin–Ciocalteu reagent, and 1.5 mL of sodium carbonate (20%, *w/w*). The mixture was incubated at room temperature for 60 min for color development. The absorbance was measured at 750 nm. The blank was prepared by replacing the extract with distilled water. Measurements were carried out in triplicates for each sample. The total phenols content in the obtained extracts was calculated by interpolating the measured sample absorbance into a calibration curve defined with standard solutions of gallic acid, defined for the concentration range 0.15–0.65 mg/mL ($A = 1.1511c − 0.0064$, $r^2 = 0.9986$). The results were expressed as gallic acid equivalents per gram of lignin material (mg GAE/g).

*2.5. Determination of Other Aromatics*

The total content of other aromatics in the obtained extracts was determined using the colorimetric assay based on the formation of a complex with aluminum. Subcritical water extracts of lignin were mixed with 5% sodium nitrite solution (0.3 mL). After 5 min, aluminum chloride hexahydrate (10%, 0.3 mL) was added and allowed to stand for 6 min. Then, sodium hydroxide (1 mol/L, 1 mL) and distilled water was added to the mixture to bring the final volume to 10 mL. The blank was prepared by replacing the extract with distilled water. The absorbance was measured at 510 nm. All determinations were repeated three times. The total aromatics content in the obtained extracts was calculated by interpolating the measured sample absorbance into the calibration curve defined with standard solutions of rutin, defined for the concentration range 0.1–0.7 mg/mL ($A = 1.1023c − 0.0084$, $r^2 = 0.9968$). The results were expressed as rutin equivalents per gram of dry lignin (mg RE/g).

*2.6. Determination of Total Antioxidant Capacity*

The total antioxidant activity of the samples was evaluated by the phosphomolybdenum method previously described by Prieto et al. [12]. Aqueous samples (0.3 mL) were

mixed with 3 mL of reagent solution (0.6 M sulfuric acid, 28 mM sodium phosphate, and 4 mM ammonium molybdate). The mixtures were incubated at 95 °C for 90 min. After cooling to room temperature, the absorbance was measured at 695 nm. The blank was prepared using methanol instead of the aqueous sample. All tests were performed in triplicates. Ascorbic acid was used as the standard, and the total antioxidant capacity was expressed as micrograms of ascorbic acid equivalents per gram of aqueous sample (μg AA/g).

## 3. Results

### 3.1. The Influence of the Catalyst

Due to the stability of the bonds between units constituting lignin, and owing to the strength of C-O bonds, catalysts are essential for increasing the efficiency of the process, or to allow the hydrothermal process in less rigorous conditions. In this investigation, lignin depolymerization with and without both homogeneous and heterogenous catalysts was investigated. The efficiency of two homogeneous (NaOH, HCl) and five heterogeneous catalysts, as indicated in Tables 1 and 2, in the depolymerization of lignin at 180 °C in subcritical water in both $N_2$ and $CO_2$ atmospheres was compared. In addition, the process without catalyst addition but with the addition of 0.5% phenol as a capping agent was compared in the two gas atmospheres (Tables 1 and 2). During hydrothermal conversion, the most easily broken bonds are C-C bonds, where lignin is broken into phenolic monomers [13]. In the presence of catalysts, the reaction products from hydrothermal conversion maintain aromatic rings while the substituent groups show different trends depending on the reaction conditions.

**Table 1.** The content of total phenols in aqueous fractions obtained by hydrothermal treatment of lignin in the presence of different catalysts.

| Total Phenols (mg GAE/g $\pm$ 2SD) | | |
|---|---|---|
| **Catalyst** | **$N_2$** | **$CO_2$** |
| Without catalyst | 35.30 $\pm$ 0.31 | 42.25 $\pm$ 0.63 |
| 0.5% Phenol | 153.09 $\pm$ 1.29 | 168.48 $\pm$0.34 |
| 1% HCl | 112.85 $\pm$ 0.27 | 122.82 $\pm$ 0.28 |
| 1% NaOH | 336.07 $\pm$ 1.05 | 240.81 $\pm$ 2.95 |
| 0.2% Zeolite | 34.52 $\pm$ 0.03 | 35.30 $\pm$ 0.41 |
| 0.2% $TiO_2$ | 35.55 $\pm$ 0.51 | 37.52 $\pm$ 0.69 |
| 0.2% $ZrO_2$ | 34.72 $\pm$ 0.70 | 45.01 $\pm$ 0.73 |
| 0.2% $CeO_2$ | 31.20 $\pm$ 0.25 | 35.44 $\pm$ 0.13 |
| 0.2% ($CeO_2$-$ZrO_2$ = 80:20) | 33.59 $\pm$ 0.88 | 34.30 $\pm$ 0.30 |

180 °C, lignin:water = 1:30, 1 h, 3 Hz.

**Table 2.** The influence of temperature on lignin depolymerization.

| T (°C) | Total Phenols $\pm$ 2SD * (mg GAE/g) | | Other Aromatics $\pm$ 2SD (mg RE/g) | |
|---|---|---|---|---|
| | **$N_2$** | **$CO_2$** | **$N_2$** | **$CO_2$** |
| 120 | 341.88 $\pm$ 0.71 | 164.88 $\pm$ 1.40 | 114.29 $\pm$ 0.54 | 58.48 $\pm$ 1.15 |
| 150 | 426.01 $\pm$ 0.35 | 249.00 $\pm$ 0.61 | 209.43 $\pm$ 4.41 | 125.21 $\pm$ 0.32 |
| 180 | 336.07 $\pm$ 1.05 | 240.81 $\pm$ 2.95 | 173.43 $\pm$ 0.80 | 103.06 $\pm$ 0.91 |
| 200 | 387.70 $\pm$ 0.36 | 317.54 $\pm$ 1.31 | 160.41 $\pm$ 0.00 | 205.67 $\pm$ 3.03 |
| 220 | 399.95 $\pm$ 2.49 | 289.31 $\pm$ 3.06 | 160.06 $\pm$ 0.37 | 193.65 $\pm$ 2.18 |

* SD—standard deviation.

### The Influence of the Catalyst Concentration

In most publications, lignin depolymerization is conducted at relatively high temperatures (250–650 °C) either with or without a catalyst [14]. The role of a catalyst is to initiate degradation reactions and increase the yields of target compounds, reduce char formation, and allow processing under milder conditions.

In our study, homogeneous catalysts, both NaOH and HCl, showed the highest efficiency and a superior performance in comparison to all tested heterogenous catalysts. In other reported studies, NaOH has also been shown to be very efficient since it increased the oil yield and limited coke formation [15]. Perez et al. used a combination of $H_2O_2$ and 1 M NaOH for rapid decomposition of lignin in a specially designed continuous flow reactor [16,17]. Aqueous $H_2O_2$ rapidly decomposed to oxygen at 400 °C, potentiating oxidative degradation of lignin. At such high temperatures and in combination with $H_2O_2$, the process of lignin depolymerization could be completed within seconds.

Heterogeneous catalysts that are often employed in hydrothermal lignin depolymerization, such as Ru-, Pt-, and Pd-based catalysts [18,19], are costly and, nowadays, are being replaced with cheaper alternatives based on transition metals, such as Cu, Ni, and Co [20]. Yamaguchi et al. [21] studied the depolymerization of oranosolv lignin in supercritical water at 400 °C, comparing the efficiency of palladium, platinum, rhodium, and ruthenium catalysts supported on charcoal. Palladium, platinum, and rhodium catalyzed lignin depolymerization into aromatic monomers, whereas with ruthenium catalyst, these reactions were not observed; however, the production of gaseous products with this catalyst (methane, carbon dioxide) was abundant.

Using lignin model molecules, benzyl phenyl ether, and diphenyl ether, the authors concluded that in supercritical water, metal catalysts depolymerize lignin by the cleavage of α-O-4, α -O-4, α-1, and 4-O-5 linkages, without the production of methoxy groups in the obtained monomers [21].

The study of Guvenatam et al. [22] showed unsatisfactory results in lignin depolymerization in supercritical water at 400 °C (4 h) with Lewis acid catalysts, i.e., metal acetates, chlorides, and triflates. In their study, the monomer yields were very low, and char was the main product that was formed. The total organic yields were between 6 and 8%, which is much lower in comparison to our study, and were achieved at much higher temperatures and longer treatment times. The authors did, however, demonstrate much better yields of alkylated monomer products in supercritical ethanol, via ethanol conversion to alkanes, alkenes, and aromatics.

Kong at al. [23] observed a remarkable increase in bio-oil yields (87.9%) with 5% nickel nitrate modified hydrotalcites. The highest monomer yields were calculated to be 35.5%, with guaiacol and guaiacol being the main products.

In addition to the demonstrated superior efficiency among all tested heterogeneous catalysts, sodium hydroxide has the advantage of being low cost and not requiring harsh reaction conditions. In our study, the influence of the sodium hydroxide concentration was investigated at 150 °C in a nitrogen atmosphere and 200 °C in a carbon dioxide atmosphere. The concentration of NaOH was varied between 0.1 and 10% and the results are shown in Figure 2.

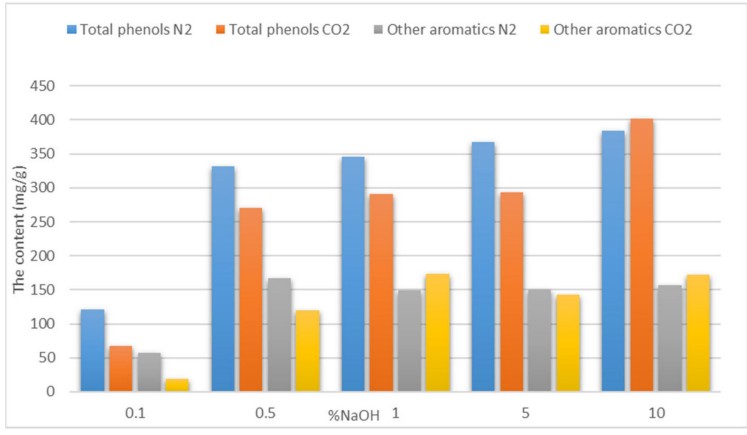

**Figure 2.** The influence of the NaOH concentration on the efficiency of lignin hydrothermal depolymerization.

### 3.2. Optimization of Lignin Depolymerization by Subcritical Water

3.2.1. Optimization of the Temperature

In this research, relatively moderate temperatures were investigated, and lignin was treated with water heated to temperatures between 120 and 220 °C in both nitrogen and carbon dioxide atmospheres. Since the addition of 1% NaOH showed remarkable effects on lignin depolymerization, this selected homogeneous catalyst was added to all tested samples. Samples were pressurized to 10 bars with inert ($N_2$) and reactive ($CO_2$) gas atmospheres and were treated within 1 h, maintaining a lignin:water ratio of 1:30. As in the case of previous experiments, higher total phenols and other aromatics were observed in lignin samples that were treated under nitrogen atmosphere, rather than a carbonic-acid-producing $CO_2$ atmosphere, which were probably partially neutralized by the added catalyst (Table 2).

3.2.2. Optimization of the Treatment Time

In lignin depolymerization, the selection of the optimal time is crucial since too long a treatment can lead to repolymerization and condensation of produced monomers, char formation, and may favor secondary reactions. In addition, shorter treatment times are related to lower energy consumption and time savings. In the used system of hydrothermal lignin, the treatment time was varied between 30 and 120 min in both nitrogen and carbon dioxide atmospheres (Table 3).

**Table 3.** The influence of time on lignin depolymerization.

| t (min) | Total Phenols (mg EGA/g) | | Other Aromatics (mg RE/g) | |
|---|---|---|---|---|
| | $N_2$ (150 °C) | $CO_2$ (200 °C) | $N_2$ (150 °C) | $CO_2$ (200 °C) |
| 30 | 349.75 ± 0.83 | 287.52 ± 1.24 | 156.97 ± 1.10 | 165.85 ± 1.45 |
| 40 | 350.13 ± 0.83 | 279.19 ± 3.31 | 156.29 ± 2.24 | 172.64 ± 2.04 |
| 60 | 426.01 ± 0.35 | 317.54 ± 1.31 | 209.43 ± 4.41 | 205.67 ± 3.03 |
| 90 | 327.25 ± 0.84 | 294.62 ± 1.71 | 140.30 ± 1.34 | 207.63 ± 3.16 |
| 120 | 332.10 ± 0.42 | 265.38 ± 3.14 | 136.87 ± 0.54 | 187.37 ± 3.89 |

Catalyst—1% NaOH; $T(N_2)$ = 150 °C; $T(CO_2)$ = 200 °C; $P(N_2)$ = 10 bar; $P(CO_2)$= 20 bar; ν = 3 Hz; r = 1:30.

3.2.3. Pressure Optimization

In previous studies on the use of subcritical water as an extracting solvent, pressure showed a negligeable effect on the extraction efficiency [24]. It should be pointed out that when using subcritical water as an extraction and reaction medium, different aspects should be considered. In the extraction of bioactive molecules from natural sources by subcritical water, it is important to maintain the integrity of the extracted compounds and avoid their decomposition. The shortest possible extraction time and an inert atmosphere are therefore required to recover target compounds with negligible loss. The pressure of the subcritical water treatment, in this case, serves to maintain water in its liquid state, having an insignificant influence on the extraction efficiency. Furthermore, an increase in pressure has been shown to cause a slight increase in the polarity of subcritical water [25]. The use of high temperatures in subcritical water extraction, in most cases, aims to cause a decrease in the water polarity in order to solubilize medium-polarity and non-polar compounds, maintaining the pressure that will keep the water in its liquid state, since the pressure has minor influence on water polarity. The use of subcritical water as a reactive medium has a different goal, and it was interesting to observe its influence on lignin decomposition in different gas atmospheres since pressurization by $CO_2$, in fact, produces carbonic acid upon gas dissolution, thus creating a more acidic environment. However, when a nitrogen inert atmosphere is used, the process is predominantly the result of water dissociation at elevated temperatures. Thus, the effect of pressure was separately investigated in nitrogen (Figure 3) and carbon dioxide (Figure 4) atmospheres, under previously adopted optimal conditions, which implied a treatment temperature of 150 °C for 1 h and 10% NaOH for

$N_2$, and a temperature of 200 °C for 1 h and 10% NaOH for $CO_2$. To both systems, 0.3% phenol was added as a capping agent.

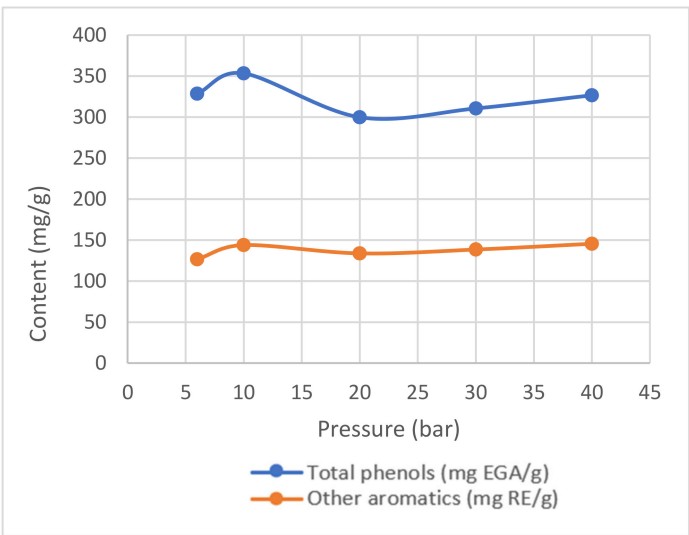

**Figure 3.** The influence of nitrogen pressure on lignin decomposition in subcritical water.

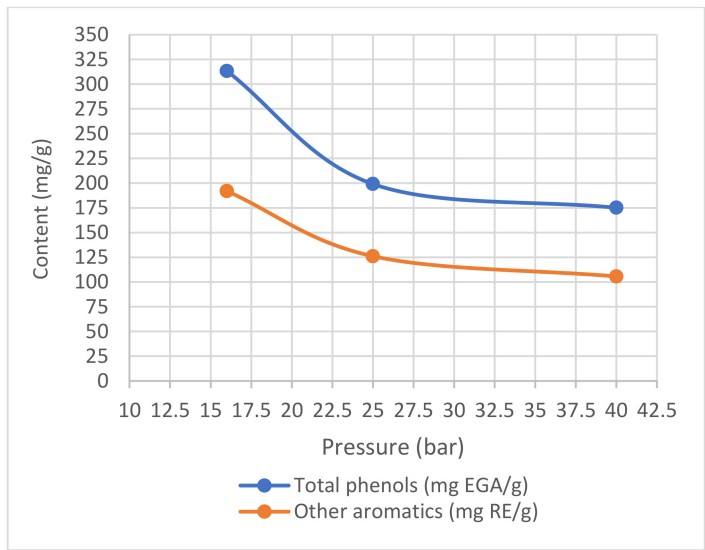

**Figure 4.** The influence of carbon dioxide pressure on lignin decomposition in subcritical water.

### 3.2.4. Capping Agent

Hydrothermal treatment of lignin, especially at higher temperatures, is accompanied by undesirable char formation, which can be minimized by the addition of capping agents. The most commonly used capping agent for lignin depolymerization is phenol. Phenol reduces char formation but also moves the equilibrium of lignin depolymerization towards the formation of phenolic monomers and other aromatics. In the study of Belkheiri et al. [6], phenol addition up to 10% did not affect the yields of bio-oil but moderately reduced char formation on the used solid catalysts ($ZrO_2$, $K_2CO_3$). Other capping agents were tested as well. In the study of Okuda et al. [26], p-cresol prevented char formation in the treatment of organosolv lignin by supercritical water (350–420 °C). The amount of cresol, however, was quite high (water:cresol = 1.8:2.5).

In our investigation, which was oriented towards maximum environmental consideration and the development of 'green' technology, we wanted to investigate changes in the process under a low concentration of capping agent: phenol. The influence of the

selected capping agent was investigated in the concentration range of up to 1%. Higher concentrations were excluded from this study due to environmental and 'green chemistry' principles, and to avoid the necessary step of phenol separation after the completion of the process. Okuda et al. [26], for example, achieved good lignin depolymerization in supercritical water at 400° without char formation; however, the ratio of phenol to water in their investigation varied between 1:1 and 5:1. The effects of the phenol concentration was observed in nitrogen and carbon dioxide atmospheres (Figures 5 and 6).

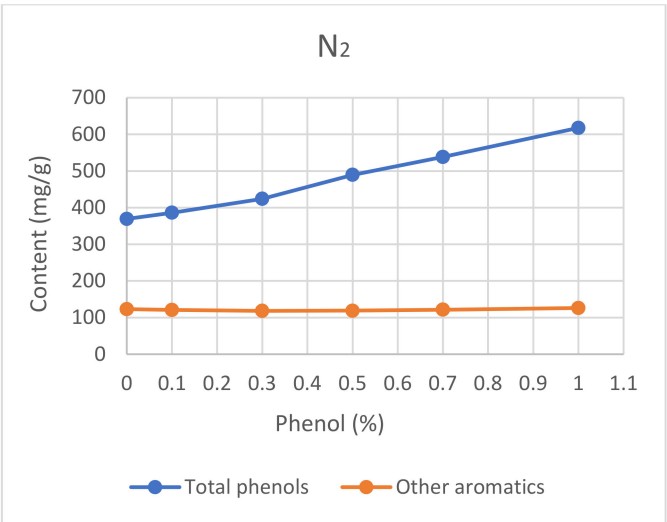

**Figure 5.** The influence of the phenol concentration on the formation of monomeric depolymerization products of lignin in a nitrogen atmosphere.

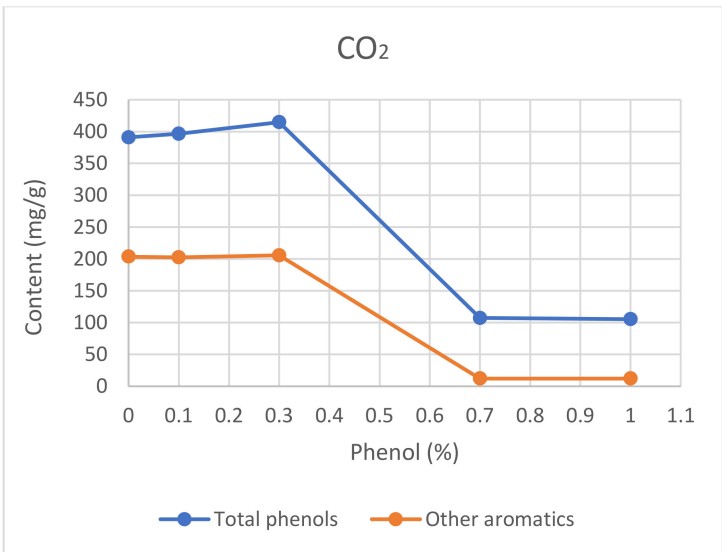

**Figure 6.** The influence of the phenol concentration on the formation of monomeric depolymerization products of lignin in a carbon dioxide atmosphere.

## 4. Total Antioxidant Activity

Phenolic compounds, such as phenolic acids and flavonoids, are widely distributed in nature, being produced by many plants as part of the protective mechanisms against pests. The production of phenolic compounds by plants is increased when the plant is subjected to stress. Namely, phenolic compounds are responsible for the antioxidant and antiradical potential of many plants. In addition, natural phenolic compounds demonstrate other bioactivities, such as anti-inflammatory, cardio-protective, neuro-protective, antimicrobial,

and other activities [27]. In the course of lignin hydrolysis, phenolic monomers are formed; so, it is expected that the obtained aqueous fractions will demonstrate an antioxidant potential. The antioxidant potential of lignin depolymerization products, obtained at different temperatures and under different gas atmospheres, was measured by the total antioxidant test and expressed as ascorbic acid equivalents (Table 4).

**Table 4.** Total antioxidant activity of lignin hydrolysates.

| t (°C) | mg AAE/g $\pm$ 2SD | |
|---|---|---|
| | $N_2$ | $CO_2$ |
| 120 | 633.69 $\pm$ 3.58 | 371.21 $\pm$ 5.39 |
| 150 | 700.42 $\pm$ 3.48 | 574.11 $\pm$ 1.72 |
| 180 | 544.44 $\pm$ 21.28 | 476.77 $\pm$ 6.36 |
| 200 | 608.18 $\pm$ 7.21 | 548.34 $\pm$ 2.28 |
| 220 | 604.82 $\pm$ 0.00 | 483.00 $\pm$ 10.61 |

AAE—ascorbic acid equivalent; SWE—conditions: catalyst—1% NaOH; T = 120–220 °C, P = 10–20 bar; t = 1 h; ν = 3 Hz; r = 1:30.

The antioxidant activity of lignin is primarily affected by its phenolic hydroxyl group content and the position of the hydroxyl groups and its average molecular weight [28].

Xiao et al. [29] performed alkali hydrolysis of lignin and studied the antioxidant activity of the thus obtained hydrolysates. The authors concluded that the DPPH scavenging ability of hydrolysates increases with the number of phenolic groups and decreases with the molecular weight of the formed fragments. At a longer hydrolysis time, the authors noted a drop in the antioxidant potential, associating it with re-polymerization. Unmodified lignin, separated from the black liquor of oil palm waste, was also characterized in respect to DPPH radical inhibition after dissolution in 2-methoxy ethanol and dimethyl sulfoxide. Both lignin solutions showed good antiradical activity, which was comparable to lignins isolated from other sources; however, the 2-methoxy ethanol solutions were more active.

In our investigation, the highest total antioxidant activity was observed in the hydrolysates obtained at 150 °C, which is in agreement with the previously reported and discussed total phenols content. Furthermore, the activity was measured in a nitrogen atmosphere, which was also proved and discussed previously. At temperatures above 150 °C, a drop in activity was observed followed by an increase in the activity starting at 200 °C. It is possible that this observed increase in antioxidant activity corresponds to neoformed compounds with antioxidant properties, which is typical for subcritical water [30,31].

## 5. Discussion

### 5.1. The Influence of the Catalyst

Phenol addition had a remarkable effect on the total phenols content in the aqueous phase; however, it did not significantly affect the content of other aromatics. Both the total phenols and total aromatics were the highest when NaOH was used for catalysis, followed by HCl. The highest contents of total phenols in subcritical water extracts of lignin were observed in systems treated with the addition of 1% NaOH in a nitrogen atmosphere (336.07 mg GAE/g). The efficiency of certain decomposition reactions in subcritical water increases in the presence of $CO_2$ via the formation of carbonic acid that dissociates, liberating H+. Since NaOH was shown to be the most efficient in the depolymerization of Kraft lignin, it was expected that both the total phenols and other aromatics would be lower in samples obtained under a $CO_2$ atmosphere, which was confirmed by the experimental results obtained in this investigation. All tested heterogenous catalysts (zeolite, $TiO_2$, $ZrO_2$, $CeO_2$, $CeO_2$-$ZrO_2$ = 80:20)) showed no significant effect on lignin treatment with subcritical water at 180 °C, producing similar total phenols and other aromatics to the treatment without catalyst. The highest observed total aromatics content (161.17 mg RE/g) is lower but comparable to the total flavonoids obtained from *Pinus radiata* bark by maceration with acetone:water 70:30 *v/v* [32] and pine (*Pinus rigida*) bark extract obtained by hot water

extraction at 100 °C [33] but higher than those obtained by maceration with 50% ethanol of apple tree residues [34], *Maclura tinctoria* L. bark water extracts [35], and knotwood (*Populus tremula*) Sohxlet methanolic extracts [36].

Phenolic monomers obtained from lignin by hydrothermal conversion have mostly been analyzed in a bio-oil fraction rather than in an aqueous extract. Rana et al. [13] determined that under subcritical water conditions (350 °C, 1 h, N2), catechol (26.76%), alkyl-guaiacols (24.83%), guaiacols (19.50%), and alkyl-catechols (17.82%) were the predominant compounds. Alkaline conditions (0.5% NaOH) in combination with $MoO_3$/SBA-15 catalyst remarkably increased the yield of bio-oil (56.4%), with catechols (42.07%) and alkyl-catechols (34.59%) being the most represented products in the alkali solutions.

In another investigation, the content of water-soluble organics (light-oil) after subcritical water treatment of lignin at 200–350 °C was in the range of 28.9–44.7% [7], being comparable with the results reported in this work, which applied much lower temperatures in comparison to the authors. The most abundant monomers were guaiacol and catechol, with the highest yields of 11.8 mg/g at 300 °C and 18.8 mg/g at 350 °C, respectively.

The Influence of the Catalyst Concentration

The obtained results indicate a steady increase in the lignin depolymerization efficiency with an increasing catalyst concentration in both gas atmospheres, reaching its maximum at the highest tested concentration (10% (*w/w* based on the lignin intake). Surprisingly, at the highest tested concentration of the catalyst, the contents of total phenols and other aromatics were slightly higher in the carbon dioxide atmosphere, whereas in all the other investigations, a much better efficiency of depolymerization was observed in the nitrogen atmosphere. However, up to the highest tested catalyst concentration, the results indicated better decomposition in the nitrogen atmosphere. A sharp increase in phenols and other aromatics in both gas atmospheres was observed at the 0.5% catalyst level, which steadily slightly increased afterwards. This leads to the assumption that with a further increase in the NaOH concentration, more depolymerization products might be formed. The use of high NaOH concentrations, however, is not recommended due to corrosion and recovery considerations.

The highest yield of the phenolic monomers was ~40% and that of the other aromatic compounds was ~17.2%, measured at the 10% NaOH levels in the carbon dioxide atmosphere at 200 °C. Similar yields, however, were observed at the much lower temperature of 150 °C in the nitrogen atmosphere, namely ~38.5% for phenolics and ~15.7% for the other aromatic, respectively.

Using the same catalyst but in much higher concentrations and a specifically designed flow-through reactor, Abdelaziz et al. [37] managed to efficiently depolymerize lignin in a time frame of only two minutes. The authors used a NaOH/lignin ratio ≈1 (*w/w*) and observed a degree of depolymerization in the temperature range of 170–250 °C. They concluded that under the tested reaction conditions, the C–O ether bonds were cleaved in the lignin inter-unit structures (β-O-4, β-β, β-1, β-5), forming monomeric (guaiacol, vanillin, apocynin, piceol) and oligomeric phenolic compounds. The authors demonstrated that the depolymerization process is NaOH-consuming and directly proportional to the reaction temperature and residence time.

In ultrafast supercritical water depolymerization of lignin using a specifically designed sudden expansion micro-reactor, the authors used 0.2 M NaOH and reaction times that were shorter than 500 ms to avoid char formation [38]. In this reactor, the optimum temperature and reaction time for the highest yields of monomers and lowest char formation (4% *w/w*) were 386 °C and 300 ms, producing a light-oil yield of 60%. In the obtained light-oil, vanillin, cresol, guaiacol, creosol, and acetovanillone comprised approximately 20% *w/w*. The total aromatic monomeric yield was 10.5 % *w/w*.

### 5.2. Optimization of Lignin Depolymerization by Subcritical Water

5.2.1. Optimization of the Temperature

The yields of total phenols and other aromatics in the nitrogen atmosphere were highest at 150 °C. After that, a moderate decline in phenolic monomers was observed, probably due to repolymerization of the liberated monomers. Treatment in the carbon dioxide atmosphere required higher temperatures for a comparable efficiency since both total phenols and other aromatics were the highest at 200 °C (Table 3). The highest reactivity conditions produced more phenolic monomers in the nitrogen atmosphere at 150 °C (426 mg GAE/g) than in the carbon dioxide atmosphere at 200 °C (317.54 mg GAE/g). Oppositely, the contents of other aromatics in the nitrogen atmosphere at 150 °C and the carbon dioxide atmosphere at 200 °C did not differ, obviously representing the maximum content of such aromatic monomers that could be reached in the tested system.

Yong and Matsumura [39] compared lignin depolymerization in subcritical (300–370 °C) and supercritical (390–450 °C) water and came to the conclusion that the mechanisms involved in lignin depolymerization change dramatically depending on the temperature and that even it is much more rapid in the supercritical region, owing to the dominant radical mechanism, the depolymerization of lignin at temperatures that are too high is not recommended due to the much more intense char formation than in subcritical water.

In the study of Islam et al. [7], the fraction of light-oil obtained by subcritical water increased with temperature from 3% at 200 °C to 14.1% at 350 °C, being significantly lower in comparison to our investigation, which was carried out at much lower temperatures. Crude oil was not affected to the same extent by the temperature increase.

Zhao et al. [40] also obtained the highest yields of phenolic monomers and oligomers at the relatively high temperature of subcritical water of 325 °C, reaching a similar yield (40.38%) to that obtained in this research, which used a much lower temperature (150 °C). The authors explained that the further decline in soluble organics after heating to 350 °C was due to simultaneous depolymerization, dehydration, condensation, and competition between these processes. Degradation reactions were dominant at lower temperatures, whereas higher temperatures potentiated condensation.

5.2.2. Optimization of the Treatment Time

Since different temperatures in nitrogen and carbon dioxide atmospheres were shown to be optimal for maximum yields, an investigation was conducted at different temperatures by heating the mixture to 150 and 200 °C for the nitrogen and carbon dioxide atmospheres, respectively. The treatment time was measured from the moment the set temperature in the reaction mixture was reached. All other conditions were the same as in previous investigations, i.e., lignin-to-water ratio was 1:10, the mixture was agitated during the treatment at a frequency of 3 Hz, and the gas pressure was 10 bars for nitrogen and 20 bars for carbon dioxide. To all reaction mixtures, 1% NaOH was added to potentiate hydrolysis. The obtained results indicate that in both gas atmospheres, the highest depolymerization was observed after 60 min of treatment. Again, higher total phenols were measured after 60 min of treatment in the nitrogen atmosphere than in the carbon dioxide atmosphere. In contrast, the other aromatics were comparable but slightly higher in the carbon dioxide atmosphere (Table 3).

In another study in a similar batch reactor with stirring, the highest concentrations of phenolic monomers and oils were observed after 80 min of reaction [41]. The authors used a much higher temperature of 300 °C and concentration of the same catalyst (4% NaOH). They also reported a decline in the content of phenolic monomers after the time peak. Dell'Orco et al. [9] applied a very short reaction time of 5 min for the treatment of lignin-rich feedstock in a batch-type reactor. However, they applied much higher temperatures than in this investigation but also came to the concussion that in comparison to systems pressurized with an inert atmosphere (Ar), and systems catalyzed with 4% KOH, pressurization with carbon dioxide produced lower yields of water-soluble organics but not the bio-crude fraction or solid residue.

### 5.2.3. Pressure Optimization

In a nitrogen atmosphere, pressure did not seem to significantly affect the process, producing almost a steady amount of monomers (Figure 3). The increase in the carbon dioxide pressure provoked an almost exponential decrease in the content of both total phenolics and other aromatics (Figure 4). This could be due to the higher concentration of formed carbonic acid and consequently formed H+, which probably caused degradation of the monomers liberated from lignin. It could be concluded that, independently of the used pressurization gas, a minimum pressure is required and recommended for the process of lignin degradation by subcritical water, which is considered an operational convenience from both the technical and economical point of views.

### 5.2.4. Capping Agent

The trend of the influence of the phenol concentration observed in the nitrogen and carbon dioxide atmospheres clearly demonstrated differences in the chemical processes that took place in these systems. The type of decomposition reactions, predominant mechanisms, and generated products are likely to substantially differ, which should be confirmed by detailed chemical analysis. In this screening phase, it was obvious that in the nitrogen atmosphere, the formation of phenolic monomers increased with the phenol concentration, which did not affect the yield of the other liberated aromatic monomers. The observed dependence of the reaction in the carbon dioxide atmosphere indicates that the phenol addition had the same effect on the formation of phenolic and other aromatic monomers. At concentrations above 0.3%, phenol caused a sharp drop in both the phenolic and other aromatic monomer contents probably due to the combined mechanism of reaction pathways, which also involved the produced monomers.

In conclusion, 1% phenol is recommended for lignin decomposition by subcritical water at 150 °C in a nitrogen atmosphere, whereas no more than 0.3% phenol is needed for efficient depolymerization of lignin at 200 °C in a carbon dioxide atmosphere.

Huang et al. [42] used ethanol as a capping agent for lignin depolymerization in supercritical ethanol. O-alkylation and C-alkylation capping reactions with ethanol efficiently hindered repolymerization, increasing the yield of alkylated mono-aromatics. This process was carried out at a relatively high temperature (380 °C) and long processing time (8 h) but produced a high yield of alkylated mono-aromatics (60–86 wt%).

Another capping agent that has been tested for lignin depolymerization in subcritical water is isopropanol [43]. The authors applied relatively high temperatures (290–335 °C) and a high pressure of 250 bar, adding $Na_2CO_3$ as a heterogenous catalyst. The addition of isopropanol reduced char formation and potentiated the formation of water-soluble organics. By NMR analysis, the authors concluded that isopropanol is a good capping agent due to the cleavage of the inter-unit linkages of lignin.

## 6. Antioxidant Activity

In our investigation, for both gas atmospheres, the measured antioxidant activity was superior in comparison to pyrolytic lignin [44]. The measured antioxidant power of lignin hydrolysates was very high, superseding the ascorbic acid equivalents multiple times in five studied varieties of *Capsicum annuum* L. [45] and fruits known for their excellent antioxidant properties [46].

## 7. Conclusions

Subcritical water treatment of Kraft lignin under moderate temperatures showed promising results for lignin utilization for different purposes. In our investigation, lignin was treated with subcritical water under mild conditions (120–220 °C).

The optimal conditions of subcritical water treatment required to obtain the maximum contents of phenolic and other aromatic monomers were investigated and defined. The highest contents of total phenols in the subcritical water fractions of the lignin treatment were observed in systems treated at 150 °C with the addition of 1% of NaOH in a nitrogen

atmosphere (336.07 mg GAE/g). Phenol addition had a remarkable effect on the total phenols content in the aqueous phase; however, it did not significantly affect the content of other aromatics.

The highest yield of phenolic monomers (~40%) and other aromatic compounds (~17.2%) was achieved in systems with the addition of 10% NaOH in a carbon dioxide atmosphere at 200 °C. Similar yields, however, were seen at a much lower temperature of 150 °C in the nitrogen atmosphere, namely ~38.5% for phenolics and ~15.7% for other aromatics, respectively.

The highest total antioxidant activity was observed in lignin hydrolysates obtained at 150 °C in the nitrogen atmosphere, which is in agreement with the phenol content. At temperatures above 200 °C, an increase in activity was observed probably due to the neo-formation of compounds with antioxidant properties, which is typical in subcritical water.

**Author Contributions:** Conceptualization, methodology, writing J.Š.-G.; Investigation, lignin treatment, analysis, T.B.-B.; Writing, editing, project administration, providing lignin samples, R.J.A.G. and T.M.S.; Writing and editing, project administration, D.Š.-M.; Synthesis of catalysts, investigation, T.I.; Synthesis of catalyst, investigation, S.B.; Investigation, Z.S. All authors have read and agreed to the published version of the manuscript.

**Funding:** This work is funded by the Science Fund of the Republic of Serbia (NanoCatalyze, Grant No 7747845) and the COST Action LignoCOST (CA17128 Pan-European network on sustainable lignin valorization).

**Data Availability Statement:** All data are presented in the manuscript. Additional information can be provided by the author upon request.

**Acknowledgments:** The contribution of COST Action LignoCOST (CA17128 Pan-European network on sustainable lignin valorization) supported by COST (European Cooperation in Science and Technology), in promoting interaction, exchange of knowledge and collaborations in the field of lignin valorization, is gratefully acknowledged. The Ministry of Education, Science and Technological Development of the Republic of Serbia project 451-03-68/2020-14/200134, is also acknowledged.

**Conflicts of Interest:** The authors declare no conflict of interest.

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
