# Peer review of "Optimization and Potentials of Kraft Lignin Hydrolysates Obtained by Subcritical Water at Moderate Temperatures"

_processes, doi:10.3390/pr10102049_

Round 1
Reviewer 1 Report
The authors did some interesting work on catalytic degradation kraft lignin to phenolic chemicals. It will be nice if the authors can investigate the reaction process under hydrogen atmosphere.
Author Response
Dear Reviewer,
Thank you very much for finding our research interesting. Processes undergoing in sub- and supercritical water with different matrixes in different gas atmospheres vary remarkably, indeed. Investigation of the decomposition reactions in different gas atmospheres is foreseen in our future work. Not only in hydrogen atmosphere, which would supposingly favor hydrogenation reactions, but also in oxygen or compressed air atmosphere to observe potentiation of oxidative degradation. It would be very useful to compare processes in subcritical water in both hydrogen and oxygen/air atmospheres and to identify major and minor degradation products, as well as thermodynamics of said processes.
Thank you once again for suggestion. The authors are grateful.
Reviewer 2 Report
The description of the introduction is too long and the structure is unclear, and the conclusion of the article also has such problems. It is recommended to describe the introduction and conclusion more precisely to facilitate the readers to read.
Line 189, ‘ZrO2 and CeO2’ should be changed to ‘ZrO2 and CeO2’.
Line 205, there should be a space between ‘1.7l’.
All tables should be standard three-line meters.
All figures are rough, and the horizontal and vertical coordinates are not detailed enough.
Author Response
Dear Reviewer,
The Authors are thankful for your suggestions. All changes made to the manuscript are visible in Track changes mode and are listed below:
- The description of the introduction is too long and the structure is unclear, and the conclusion of the article also has such problems. It is recommended to describe the introduction and conclusion more precisely to facilitate the readers to read.
The Introduction and Conclusion parts were modified and made clearer and shorter to facilitate understanding of the described topic. Consequently, the reference list was modified too.
- Line 189, ‘ZrO2 and CeO2’ should be changed to ‘ZrO2and CeO2’:
The formulae of the catalysts have been corrected in subscript to ZrO2 and CeO2
- Line 205, there should be a space between ‘1.7l’:
The space was introduced between the numerical value and the liter.
- All tables should be standard three-line meters:
All tables in the manuscript were reformatted to three-line view.
- All figures are rough, and the horizontal and vertical coordinates are not detailed enough:
In Figures 3-6 axes were modified, improving the resolution and allowing more accurate reading

Round 2
Reviewer 2 Report
This version has made appropriate revisions to the corresponding revision comments, and the text quality has been greatly improved.